# Hepatic TET3 contributes to type-2 diabetes by inducing the HNF4α fetal isoform

Da Li[1,2,10], Tiefeng Cao[1,3,10], Xiaoli Sun[1,4,10], Sungho Jin[5,6], Di Xie[1,7], Xinmei Huang[1,8], Xiaoyong Yang [6], Gordon G. Carmichael [9], Hugh S. Taylor[1], Sabrina Diano [5,6] & Yingqun Huang [1*]

Precise control of hepatic glucose production (HGP) is pivotal to maintain systemic glucose homeostasis. HNF4α functions to stimulate transcription of key gluconeogenic genes. *HNF4α* harbors two promoters (P2 and P1) thought to be primarily active in fetal and adult livers, respectively. Here we report that the fetal version of HNF4α is required for HGP in the adult liver. This isoform is acutely induced upon fasting and chronically increased in type-2 diabetes (T2D). P2 isoform induction occurs in response to glucagon-stimulated upregulation of TET3, not previously shown to be involved in HGP. TET3 is recruited to the P2 promoter by FOXA2, leading to promoter demethylation and increased transcription. While TET3 over-expression augments HGP, knockdown of either TET3 or the P2 isoform alone in the liver improves glucose homeostasis in dietary and genetic mouse models of T2D. These studies unmask an unanticipated, conserved regulatory mechanism in HGP and offer potential therapeutic targets for T2D.

[1] Department of Obstetrics, Gynecology & Reproductive Sciences, Yale University School of Medicine, New Haven, CT 06510, USA. [2] Center of Reproductive Medicine, Department of Obstetrics and Gynecology, Shengjing Hospital of China Medical University, Shenyang 110004, China. [3] Department of Obstetrics and Gynecology, The First Affiliated Hospital, Sun Yat-Sen University, Guangzhou, Guangdong 510070, China. [4] Center of Reproductive Medicine, Department of Obstetrics and Gynecology, Affiliated Hospital of Nantong University, Jiangsu 226001, China. [5] Departments of Cellular and Molecular Physiology and of Neuroscience, Yale University School of Medicine, New Haven, CT 06520, USA. [6] Department of Comparative Medicine, Yale University School of Medicine, New Haven, CT 06520, USA. [7] Center of Reproductive Medicine, Department of Obstetrics and Gynecology, General Hospital of Central Theater Command, Wuhan, Hubei 430070, China. [8] Department of Endocrinology, Fifth People's Hospital of Shanghai, Fudan University School of Medicine, Shanghai 200080, China. [9] Department of Genetics and Genome Sciences, University of Connecticut Health Center, Farmington, CT 06030, USA. [10]These authors contributed equally: Da Li, Tiefeng Cao, Xiaoli Sun. *email: Yingqun.huang@yale.edu

The evolutionarily conserved, liver-enriched transcription factor HNF4α has been extensively studied for its role in hepatic differentiation and function[1]. During fasting, hepatic expression of HNF4α and its coactivator PGC-1α both increase and transcriptionally activate the rate-limiting enzymes PEPCK and G6PC (encoded by *PCK1* and *G6PC*, respectively), leading to gluconeogenesis[2–4]. *HNF4α* contains two promoters, P2 and its downstream P1, which drive multiple HNF4α isoforms (α1–α9) via alternative splicing in a development-specific and tissue-specific manner (Supplementary Fig. 1)[5,6]. The P2-derived isoforms differ from P1-derived isoforms only in their N-terminal regions. The current dogma is that the P2 isoform predominates during fetal development, however after birth the P1 isoform takes over, directing a wide range of liver functions including gluconeogenesis[5,6]. While many studies have focused on the P2-to-P1 promoter switch during hepatic differentiation[5–7], none has yet documented P2 promoter reactivation in the adult liver.

DNA methylation involves conversion of unmodified cytosine to 5-methylcytosine (5mC) by DNA methyltransferases. The Ten-eleven translocation (TET) family proteins (TET1, TET2, and TET3) facilitate active (replication-independent) and passive (replication-dependent) DNA demethylation via iteratively oxidizing 5mC to 5-hydroxymethylcytosines (5hmC), 5-formylcytosine (5fC), and 5-carboxylcytosine (5caC), followed by excision of 5fC or 5caC by thymine DNA glycosylase coupled with base excision repair[8,9]. The enzymatic activity of TETs is regulated by co-factors including α-ketoglutarate (α-KG) generated through the TCA cycle and vitamin C, and by post-translational modifications[8,9]. Besides being the key intermediate for TET-mediated DNA demethylation, 5hmC is a potential epigenetic mark; yet, the exact rules that govern how and when TETs proceed beyond 5hmC leading to DNA demethylation are unclear. Recently, non-catalytic functions of TETs have also been described[10–13]. Mechanistically, TETs as general DNA-binding proteins appear to be recruited to specific gene loci by transcription factors. For example, during B cell differentiation TET2 and TET3 cooperate with PU.1 to regulate gene expression[14]. PU.1 is a pioneer transcription factor (PTF), defined by its ability to directly associate with target sequences on condensed chromatin opening the local chromatin for other factors to bind[9,15]. Among the numerous loci with which PU.1 associated was the Igκ enhancer. Using a TET2/TET3 double knockout (KO) mouse model and TET/PU.1 co-immunoprecipitation (co-IP) and PU.1 shRNA knockdown in B cells, it was demonstrated that PU.1 interacts with and guides TET2/TET3 to the Igκ enhancer for 5hmC deposition and subsequent demethylation[9].

While much is known about the role of TETs in development, stem cells, and cancer[8,9,16], little is known about their role in energy metabolism. In the current study we report an unexpected finding of P2 promoter reactivation in adult liver by TET3 with an essential role in control of hepatic glucose production (HGP).

## Results

**Fasting upregulates hepatic TET3.** We have previously documented that fasting upregulates hepatic expression of H19 long noncoding RNA contributing to increased expression of *HNF4α*[17]. Overnight fasting expectedly increased levels of H19 as well as mRNAs for PGC-1α, HNF4α, PEPCK, and G6PC; curiously, fasting also increased mRNA expression of TET3, but not of its family members TET2 and TET1 (Supplementary Fig. 2a). To determine how TET3 is upregulated, primary hepatocytes from wild-type (WT) and H19 KO mice[17,18] were stimulated with glucagon. In WT hepatocytes, *H19* expression was readily induced by glucagon, as was *TET3*; however, in KO hepatocytes, glucagon no longer stimulated *TET3* expression (Supplementary Fig. 2b). Next, H19 was expressed in WT primary hepatocytes

with an adeno-associated virus-based vector (AAV-H19[17]) in absence of glucagon. Exogenous H19 expression increased TET3 mRNA levels (Supplementary Fig. 2c). Consistently, livers from ad libitum-fed mice injected with AAV-H19 showed increased TET3 mRNA (Supplementary Fig. 2d). H19 contains multiple microRNA let-7-binding sites, acting as a molecular sponge for let-7[19]. Recently, we identified TET3 as a target of let-7-mediated repression of expression at the posttranscriptional level (both human and mouse TET3 mRNAs contain let-7-binding sites) and demonstrated that H19 promotes TET3 expression by reducing the bioavailability of let-7[20]. Thus, H19 enables glucagon-induced TET3 upregulation, likely via inhibiting let-7, in isolated hepatocytes.

**TET3 promotes HGP.** To determine whether *TET3* expression in the absence of upstream stimulators (glucagon and H19) is sufficient to enhance glucose production, TET3 was expressed in primary hepatocytes from H19 KO mice. Hepatocytes were infected with viruses containing a cDNA encoding TET3 (Ad-TET3) or green fluorescent protein (Ad-GFP). When TET3 was overexpressed, increased expression of *PCK1* and *G6PC* was evident (Fig. 1a, b). TET3 overexpression also increased glucose production (Fig. 1c). In contrast, when WT hepatocytes were infected with AAV-siTET3 (specific against mouse TET3, or a non-targeting siRNA control AAV-scr) in the presence of glucagon stimulation, it led to decreased expression of *PCK1* and *G6PC* (Fig. 1d, e) and glucose production (Fig. 1f). TET3 knockdown did not affect the expression of TET2 and TET1 (Supplementary Fig. 3a), confirming the specificity of the TET3 siRNA. As stated in our "Methods" section, all primary hepatocyte experiments (e.g., glucagon stimulation and TET3 overexpression) were performed on cells maintained in a complete culture medium (CM) containing serum, insulin, and dexamethasone, conditions optimized and important for cell viability[17]. These conditions allowed cell viability to persist to the end of the experiments (Supplementary Fig. 3b). Our additional rationale for performing glucagon stimulation in the presence of insulin was derived from the fact that insulin is present in the circulation during fasting, albeit at a lower level as compared to fed conditions. Taken together, our results show that TET3 augments glucose production, at least in part by increasing expression of key gluconeogenic genes in isolated hepatocytes.

To demonstrate that TET3 regulates HGP in vivo, Ad-TET3 (or Ad-GFP) viruses were infused through the tail vein into H19 KO mice. Systemic administration of recombinant adenoviruses into rodents resulted in expression of transgenes in the liver, with no detectable expression in other peripheral tissues and brain; nor was hepatotoxicity detected[2,21,22]. Following 10 days of viral administration, mice fed ad libitum were sacrificed and liver and blood samples were harvested. Mice infused with Ad-TET3 had a significant increase in hepatic TET3 expression relative to mice infused with Ad-GFP, which was accompanied by increased expression of *PCK1* and *G6PC* (Fig. 1g, h); there was also an increase in blood glucose and insulin levels (Fig. 1i).

To determine whether upregulation of TET3 is necessary for HGP during fasting, AAV-siTET3 or AAV-scr were injected via tail vein into WT mice followed by fasting 10 days later. Systemic infusion of recombinant AAVs into mice leads to liver-specific expression of transgenes[17]. Mice infused with AAV-siTET3 showed a significant decrease in fasting blood glucose and fasting insulin, as compared to AAV-scr infused animals (Fig. 1j). Pyruvate tolerance tests (PTT, a readout for HGP) showed lower glucose levels following pyruvate injection (Fig. 1k). Protein analyses revealed decreased levels of TET3, PEPCK, and G6PC in

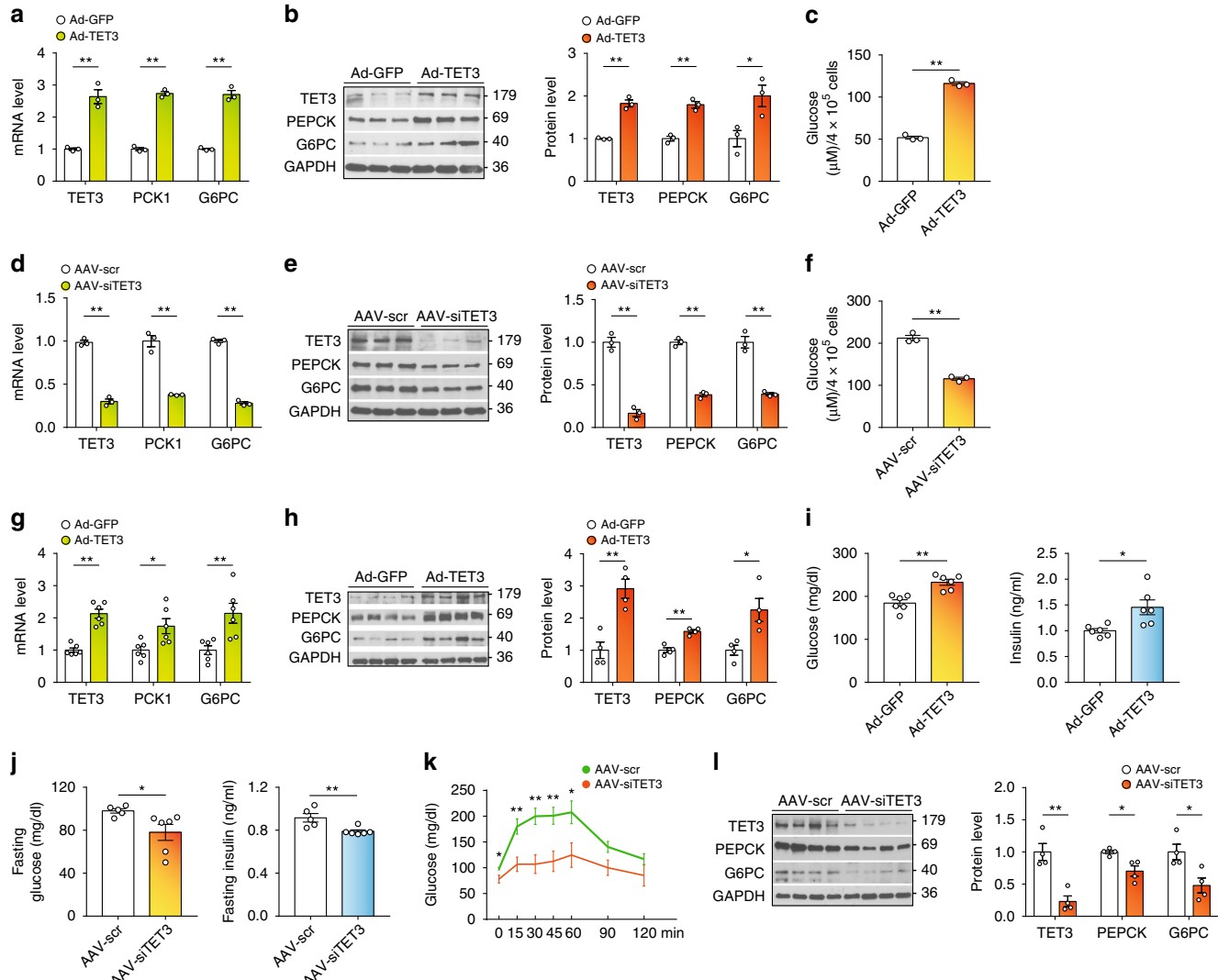

**Fig. 1 TET3 promotes HGP. a** and **b** qPCR and immunoblotting (IB) of TET3, PCK1, and G6PC at 72 h following infection with Ad-GFP or Ad-TET3 in H19 KO hepatocytes. $n = 3$. Molecular sizes of the proteins in kDa are marked on the right. **c** Glucose production by H19 KO hepatocytes at 72 h following infection with Ad-GFP or Ad-TET3. $n = 3$. **d** and **e** qPCR and IB of TET3, PCK1, and G6PC at 48 h of glucagon stimulation following infection with AAV-scr or AAV-siTET3 for 24 h in primary hepatocytes. $n = 3$. **f** Glucose production by primary hepatocytes at 48 h of glucagon stimulation following infection with AAV-scr or AAV-siTET3 for 24 h. $n = 3$. **g** and **h** qPCR and IB of TET3, PCK1, and G6PC from liver tissues isolated from H19 KO mice injected via tail vein with Ad-GFP or Ad-TET3. $n = 4$–6. Liver tissues were collected from ad libitum-fed mice on day 10 post infection. **i** Blood glucose and insulin levels of H19 KO mice treated as in **g**. $n = 6$. **j** Fasting blood glucose and insulin levels of mice 10 days after infection with AAV-scr or AAV-siTET3. $n = 5$–6. **k** PTT in mice treated as in **j**. $n = 5$–6, two-way ANOVA with Sidak post-test. **l** IB of TET3, PEPCK, and G6PC proteins from liver tissues isolated from mice treated as in **j**. $n = 4$. Data **a**–**f** are representative of at least two independent experiments; data **g**–**l** are representative of two independent experiments. Two-tailed Student's *t* tests (or as otherwise indicated) were used to compare means between groups. All data are presented as mean ± SEM. *$p < 0.05$, **$p < 0.01$. Source data are provided as a Source Data File.

livers of AAV-siTET3 relative to AAV-scr-injected animals (Fig. 1l). Based on these results we conclude that TET3 is a regulator of HGP.

**TET3 reactivates the P2 promoter.** Increased H19 expression was detected in the liver during fasting and in livers of human and mouse with type-2 diabetes (T2D)[17,23], conditions known to have physiological and pathological increase in gluconeogenesis, respectively. Consistent with the notion that H19 positively regulates *TET3* expression[20], elevated TET3 was evident in all conditions (Fig. 2a–d), where H19 expression was increased[17]. Importantly, mining of human liver databases[24,25] revealed a significant increase in expression of *TET3* in the liver of T2D patients as compared to non-diabetic controls (Supplementary

Fig. 3c). Together, these results suggest that the H19/TET3-mediated regulation of HGP is likely conserved between human and mouse.

Remarkably, conditions that resulted in increased *TET3* expression also led to an increase in expression of the P2- (but not P1) specific HNF4α isoform at levels of both mRNA (Fig. 2e–h) and protein (Fig. 2i–l). TET3 overexpression (Fig. 2m) increased P2 isoform both at the mRNA (Fig. 2n) and protein (Fig. 2o) levels without affecting P1 isoform levels. In contrast, TET3 knockdown (Fig. 2p) selectively decreased the P2 isoform (Fig. 2q, r). These data led to our hypothesis that TET3-mediated reactivation of HNF4α P2 promoter and the derived isoform may reflect a previously unexpected mechanism of gluconeogenesis activation in adult liver.

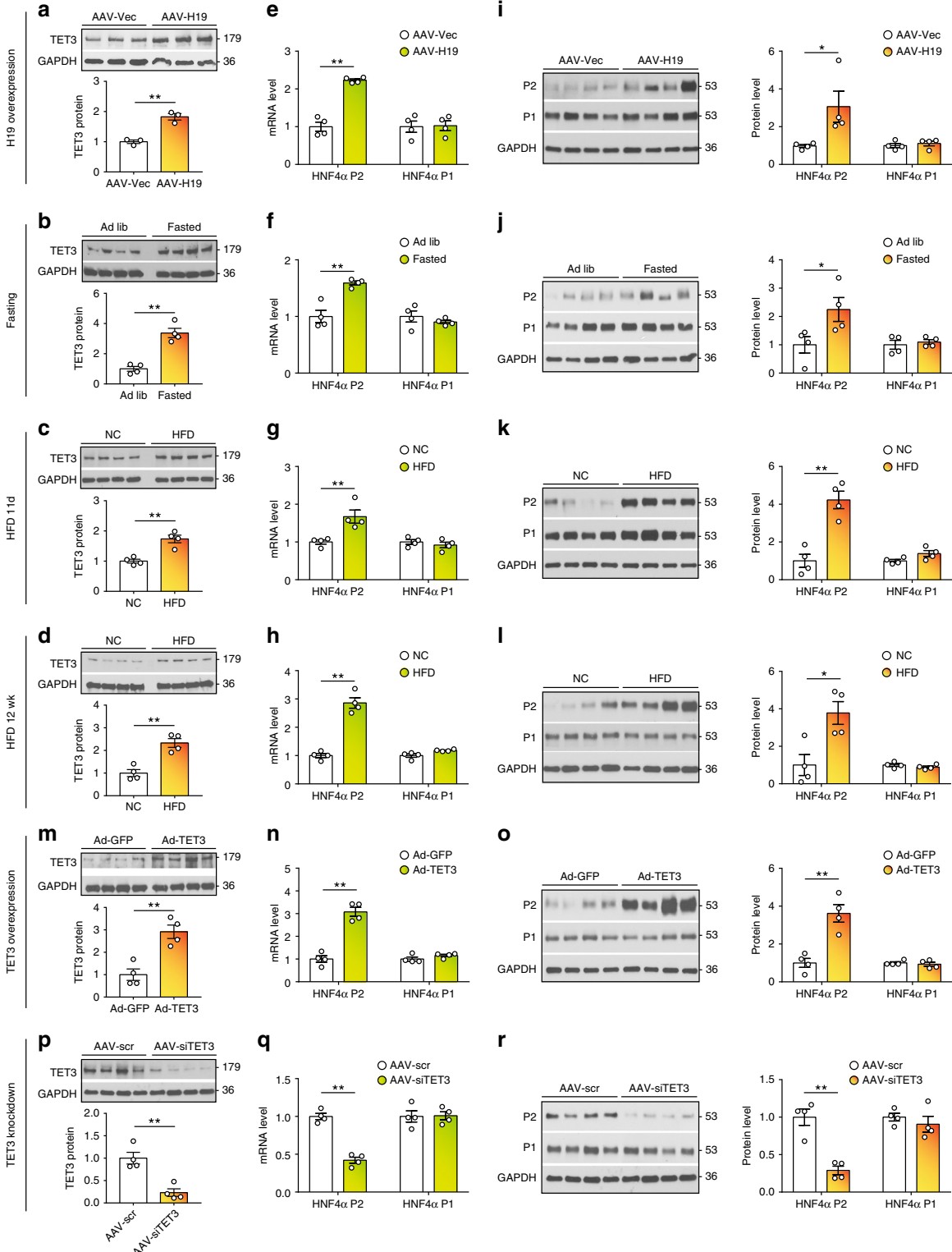

**Fig. 2 TET3 expression positively correlates with P2 isoform expression. a–d** IB of TET3 from liver tissues isolated from mice infected with AAV-Vec/ AAV-H19, ad libitium fed/fasted, exposed to normal chow (NC)/HFD for 11 days or 12 weeks. $n = 3$–4. **e–h** qPCR of HNF4α P2 and P1 isoforms in liver tissues from mice under the indicated conditions. $n = 4$. **i–l** IB of HNF4α P2 and P1 proteins in liver tissues from mice under the indicated conditions. $n = 4$. **m** IB of TET3 in liver tissues from H19 KO mice infected with Ad-GFP or Ad-TET3 for 10 days. $n = 4$. **n** and **o** qPCR and IB of HNF4α P2 and P1 isoforms in liver tissues from H19 KO mice treated as in **m**. $n = 4$. **p** IB of TET3 in liver tissues from mice infected with AAV-scr or AAV-siTET3 for 10 days. $n = 4$. **q** and **r** qPCR and IB of HNF4α P2 and P1 isoforms in liver tissues from mice treated as in **p**. $n = 4$. All data are representative of at least two independent experiments. Two-tailed Student's $t$ tests were used and data are presented as mean ± SEM. *$p < 0.05$, **$p < 0.01$. Source data are provided as a Source Data File.

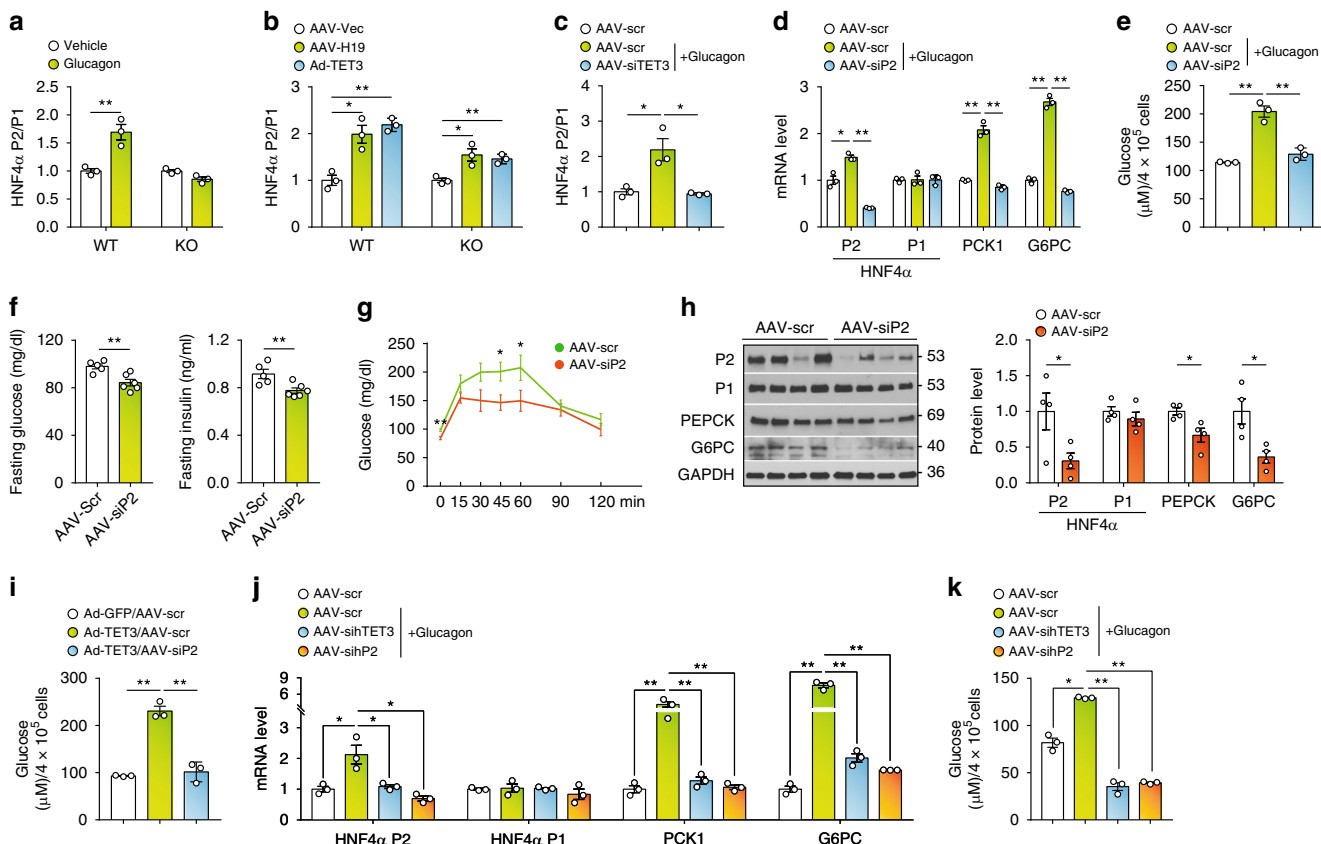

**Fig. 3 The mechanism of TET3-induced P2 isoform is conserved. a** qPCR of HNF4α isoforms from WT and H19 KO hepatocytes treated with glucagon (20 nM) or vehicle for 24 h. The ratio of P2-to-P1 isoforms is arbitrarily set as 1 in vehicle-treated group. $n = 3$, Two-tailed Student's $t$ tests. **b** qPCR of HNF4α isoforms from hepatocytes infected with indicated viruses for 48 h. $n = 3$, One-way ANOVA with Dunnett post-test. **c** qPCR of P2 and P1 isoforms from primary hepatocytes infected with AAV-scr or AAV-siTET3 for 48 h and treated with vehicle or glucagon for 24 h. $n = 3$, One-way ANOVA with Tukey post-test. **d** qPCR of HNF4α isoforms and PCK1 and G6PC from primary hepatocytes infected with AAV-scr or AAV-siP2 for 48 h, followed by treatment with glucagon. HNF4α isoforms were analyzed after 24 h of glucagon stimulation; PCK1 and G6PC were analyzed after 48 h of glucagon stimulation. $n = 3$, One-way ANOVA with Tukey post-test. **e** Glucose production from primary hepatocytes infected with AAV-scr or AAV-siP for 48 h and treated with vehicle or glucagon for 48 h. $n = 3$, One-way ANOVA with Tukey post-test. **f** Fasting blood glucose and insulin levels of mice infected with AAV-scr or AAV-siP2 for 10 days. $n = 5$–6, Two-tailed Student's $t$ tests. **g** PTT in mice treated as in **f**. $n = 5$–6, Two-way ANOVA with Sidak post-test. **h** IB of indicated proteins from liver tissues isolated from mice treated as in **f**. $n = 4$, Two-tailed Student's $t$ tests. **i** Glucose production from H19 KO hepatocytes at 72 h following infection with indicated viruses. $n = 3$, One-way ANOVA with Tukey post-test. **j** qPCR of indicated genes in primary human hepatocytes infected with AAV-scr, sihTET3, or sihP2 for 48 h and treated with glucagon or vehicle for 24 h. $n = 3$, One-way ANOVA with Tukey post-test. **k** Glucose production from primary human hepatocytes treated as in **j**. $n = 3$, One-way ANOVA with Tukey post-test. Data **a**–**e** are representative of at least two independent experiments; data **f**–**k** are representative of two independent experiments. All data are presented as mean ± SEM. *$p < 0.05$, **$p < 0.01$. Source data are provided as a Source Data File.

**The P2 isoform mediates TET3-dependent HGP**. Next, we used primary mouse hepatocytes to test effects of glucagon and H19 on P2 and P1 promoter usage. While glucagon increased P2 usage in WT hepatocytes, it failed to do so in H19 KO hepatocytes (Fig. 3a). On the other hand, without glucagon exogeneous expression of either H19 or TET3 alone was sufficient to increase P2 usage in both WT and KO hepatocytes (Fig. 3b). Further, when TET3 was downregulated by siRNA, glucagon no longer induced P2 (Fig. 3c). Together, these results show that glucagon selectively induces the P2 isoform and that TET3 is required for this induction to occur in hepatocytes.

To address whether an increase in the P2 isoform is necessary for enhanced glucose production, we tested effects of P2-specific siRNA (AAV-siP2) knockdown on gluconeogenic gene expression. As seen in Fig. 3d, glucagon expectedly increased the P2 (but not P1) isoform in hepatocytes; it also increased expression of *PCK1* and *G6PC*. Infection with AAV-siP2 downregulated the P2 isoform without affecting the P1 isoform (Fig. 3d). Importantly, knockdown of the P2

isoform not only decreased *PCK1* and *G6PC* expression (Fig. 3d), but also abolished glucagon-induced glucose production (Fig. 3e). Thus, the P2 isoform is critical for glucagon-induced glucose production in isolated hepatocytes. To determine whether this regulation also occurs in vivo, mice were injected with AAV-scr or AAV-siP2 viruses. Ten days later, mice injected with AAV-siP2 showed decreased fasting glucose and fasting insulin (Fig. 3f) and decreased PTT (Fig. 3g), as compared to AAV-scr-injected animals. Protein analysis confirmed selective decrease in the P2 isoform, with a concomitant decrease in PEPCK and G6PC (Fig. 3h). To address whether HNF4α P2 is required for TET3-induced glucose production, TET3 was overexpressed in H19 KO hepatocytes in the presence of HNF4α P2 knockdown. P2 knockdown (Supplementary Fig. 3d) abolished TET3-induced glucose production (Fig. 3i), suggesting that HNF4α P2 is the major effector of TET3-mediated regulation of HGP.

Next, primary human hepatocytes were infected with AAV-scr, AAV-sihTET3 (siRNA against human TET3), or AAV-sihP2

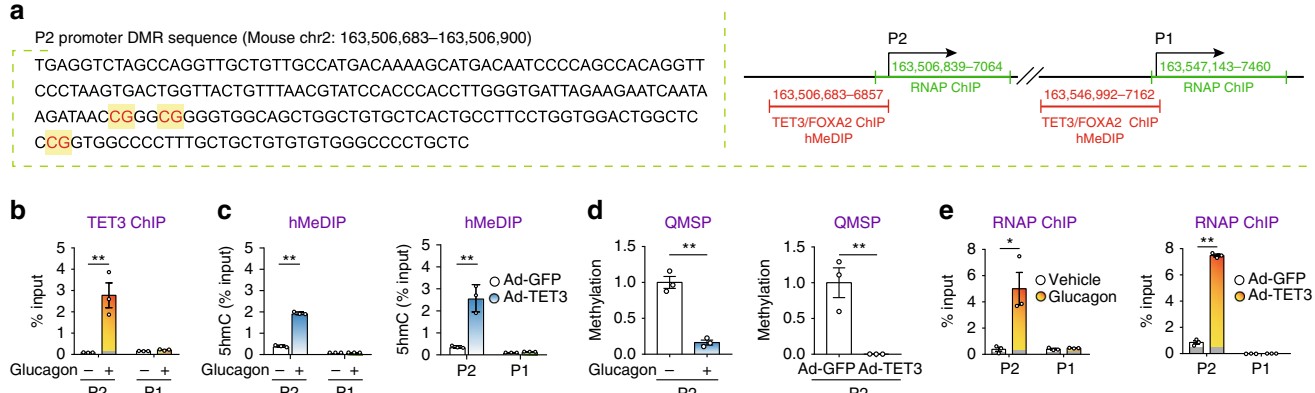

**Fig. 4 TET3 induces epigenetic modification at the P2 promoter. a** Left, DMR sequence in the P2 promoter of mouse *Hnf4α*. The three differentially methylated CpGs are highlighted. Numbers mark the starting and ending positions of nucleotides in the chromosome. Right, schematic of mouse *Hnf4α* showing TET3/FOXA2 ChIP and hMeDIP (red) and RNAP ChIP (green) regions, respectively. Numbers mark the starting and ending positions of nucleotides in the chromosome. Not drawn to scale. **b** Mouse primary hepatocytes were treated with vehicle (−) or 20 nM of glucagon (+) for 24 h, followed by ChIP-qPCR of P2 and P1 promoters. Data are presented as % input, with gray bars representing background IgG signal. *n* = 3. **c** Left, mouse hepatocytes were treated as in **b**, followed by hMeDIP of P2 and P1 promoters. *n* = 3. Right, mouse hepatocytes were infected with Ad-GFP or Ad-TET3, followed by hMeDIP at 36 h post-infection. *n* = 3. **d** Left, mouse hepatocytes were treated as in **b**, followed by QMSP of P2. Right, H19 KO hepatocytes were infected with Ad-GFP or Ad-TET3, followed by QMSP of P2 at 48 h post-infection. *n* = 3. **e** Left, mouse hepatocytes were treated with vehicle or glucagon for 24 h, followed by RNAP ChIP of P2 and P1 promoters. Right, H19 KO mouse hepatocytes were infected with Ad-GFP or Ad-TET3, followed by RNAP ChIP at 48 h post-infection. *n* = 3. Data **b**–**e** are representative of two independent experiments and are presented as mean ± SEM, using Two-tailed Student's *t* tests. *$p < 0.05$, **$p < 0.01$.

(siRNA specific for human HNF4α P2) and treated with vehicle or glucagon. Glucagon stimulated expression from P2 (but not P1) and also increased expression of *PCK1* and *G6PC*; knockdown of either TET3 or the P2 isoform was sufficient to reduce PCK1 and G6PC mRNAs to the control levels (Fig. 3j). Knockdown of TET3 or the HNF4α P2 isoform also led to decreased glucose production (Fig. 3k). The lower than basal level of glucose production in the TET3 or P2 isoform knockdown group reflected persistent siRNA effects. Neither *TET1* nor *TET2* expression was affected by AAV-sihTET3 (Supplementary Fig. 3e) and the P1 isoform level was not altered by AAV-sihP2 (Fig. 3j). Based on these data the mechanism of TET3-mediated HNF4α P2 isoform induction in HGP is likely conserved between human and mouse.

**TET3 promotes demethylation of the P2 promoter.** We previously reported a positive correlation between increased H19 expression and *HNF4α* promoter hypomethylation in human hepatoma cells and in livers of multiple mouse models, including fasting, high fat diet (HFD)-induced T2D, and liver-specific H19 overexpression[17]. The differentially methylated region (DMR) mapped to the *HNF4α* P2 promoter known to be highly conserved between human and mouse (Fig. 4a, left)[17]. As *TET3* expression was also increased under these conditions (Fig. 2a–d), we speculated that TET3 may activate P2-specific transcription by binding and demethylating the P2 promoter. TETs bind and demethylate DNA leading to transcription activation[16,20,26,27].

First, we tested whether glucagon promotes TET3 binding to the P2 promoter. Mouse hepatocytes were treated with glucagon or vehicle, followed by chromatin immunoprecipitation coupled with qPCR (ChIP–qPCR). We used a TET3-specific antibody[20] to immunoprecipitate protein–DNA complexes from hepatocytes and qPCR amplified the P2 and P1 promoters (Fig. 4a, right). Glucagon stimulation dramatically increased TET3 binding to the P2 (but not P1) promoter (Fig. 4b). Next, we determined effects of glucagon and TET3 on promoter 5hmC deposition using hydroxymethylated DNA immunoprecipitation-qPCR (hMeDIP-qPCR). Mouse hepatocytes were treated with glucagon (vehicle as

control) or infected with Ad-TET3 (Ad-GFP as control). Either glucagon stimulation or exogenous TET3 expression increased 5hmC at the P2 (but not P1) promoter (Fig. 4c), demonstrating that glucagon promotes direct binding of TET3 to the P2 promoter inducing 5hmC deposition. To address whether TET3 binding facilitates promoter demethylation, quantitative methylation-specific PCR (QMSP) was performed using previously described methods[20,28,29]. The QMSP primers were designed based on the differentially methylated cytosine residues within the DMR of the P2 promoter (Fig. 4a, left). Figure 4d shows that treatment of hepatocytes with glucagon or with exogenous TET3 expression significantly decreased methylation at the P2 promoter without affecting the P1 promoter. Next, we performed ChIP assays using anti-Ser-5(P)-RNP antibody that specifically recognizes activated RNA polymerase (RNAP)[28]. We compared binding of activated RNAP to *HNF4α* between glucagon and vehicle-treated hepatocytes to assess relative transcription activity of both P2 and P1 promoters (Fig. 4a, right). We observed an ~5-fold enrichment in RNAP at the P2 promoter in glucagon-stimulated versus vehicle-treated hepatocytes and no change in RNAP association with P1 (Fig. 4e). Similar results were obtained when TET3 was overexpressed in the absence of glucagon stimulation (Fig. 4e). Collectively, our results demonstrate that TET3 activates P2-specific transcription by directly binding and demethylating the P2 promoter.

**TET3 functionally and physically interacts with FOXA2.** FOXA2 is essential for liver development and function via acting as a PTF[15,30]. Using ChIP-seq FOXA2 was found to bind 45% of expressed genes in the adult mouse liver, acting either as an inducer or repressor of gene expression depending on its binding location[31]. FOXA2 is required for fasting-induced hepatic gluconeogenesis in part by enabling recruitment of cAMP response element-binding protein (CREB) and the glucocorticoid receptor (GR) to their respective target genes[32]. In the mouse liver FOXA2 occupies its target sites constitutively regardless of metabolic state (i.e., fasted or fed), yet binding of activated CREB and GR (respectively, by glucagon and glucocorticoids) to their target

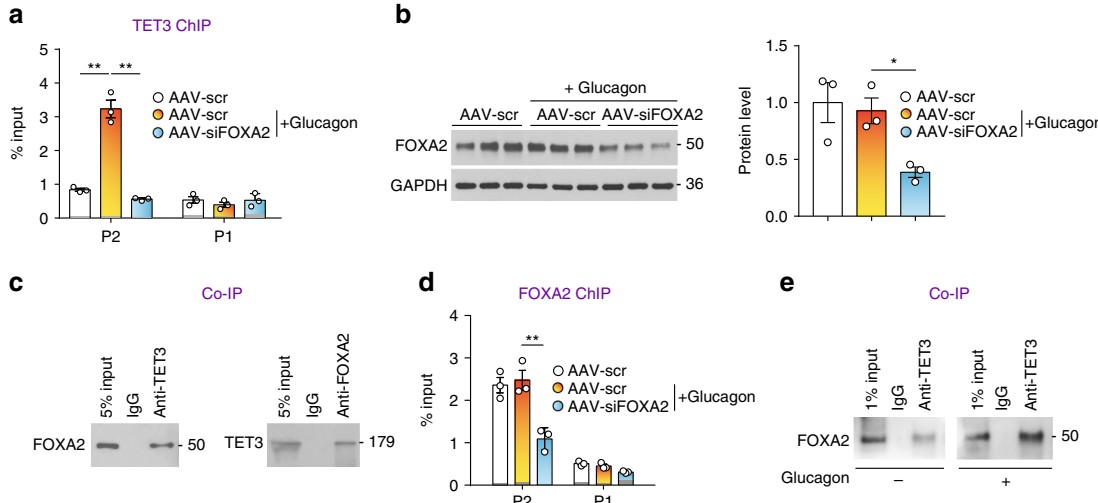

**Fig. 5 TET3 interacts with FOXA2. a** TET3 ChIP-qPCR of mouse primary hepatocytes infected with AAV-scr or AAV-siFOXA2 for 48 h, followed by glucagon (20 nM) stimulation for 48 h. $n = 3$, one-way ANOVA with Tukey post-test. **b** IB of FOXA2 of mouse primary hepatocytes infected with AAV-scr or AAV-siFOXA2 for 48 h, followed by glucagon stimulation for 48 h. $n = 3$, one-way ANOVA with Tukey post-test. **c** IB of immunoprecipitated FOXA2 and TET3 from H19 KO hepatocytes infected with Ad-TET3 for 24 h, showing that FOXA2 and TET3 are present in each other's complexes. The secondary antibodies used were Rabbit IgG TrueBlot® and Goat IgG TrueBlot®, respectively. These HRP-conjugated monoclonal secondary antibodies enable detection of immunoblotted target proteins without hindrance by interfering immunoprecipitating immunoglobulin heavy and light chains. **d** FOXA2 ChIP-qPCR of mouse primary hepatocytes treated as in **a**. $n = 3$, one-way ANOVA with Tukey post-test. **e** Mouse primary hepatocytes were treated with glucagon (+) or vehicle (−) and co-IP using TET3 antibody was carried out 24 h later. IB results show increased association of FOXA2 with TET3 following glucagon stimulation. All data are representative of two independent experiments and are presented as mean ± SEM. $*p < 0.05$, $**p < 0.01$. Source data are provided as a Source Data File.

genes was significantly enhanced by fasting (based on ChIP-qPCR) and the effect was abolished in FOXA2-deficient hepatocytes[32]. Given the proximity of FOXA2-binding sites to those recognized by activated CREB and GR on a subset of gluconeogenic genes, it has been proposed that binding of FOXA2 to its *cis*-acting elements opens the local chromatin structure thereby facilitating binding of other transcription factors such as CREB and GR[32]. Because PU.1 (a PTF) interacts with and recruits TET2/TET3 to the Igκ enhancer[9] and because FOXA2 is a PTF required for activation of the hepatic gluconeogenic transcriptional program in response to fasting, we hypothesized that FOXA2 may play a role in TET3-dependent P2 promoter reactivation.

The first evidence supporting our hypothesis was the observation of the preferential enrichment of FOXA2 at the P2 versus P1 promoters of *HNF4α* in the human liver (Supplementary Fig. 4, and ENCODE ChIP-seq GSE105237 and GSE105487). This preferential binding of FOXA2 to the P2 promoter has not been previously reported in the literature. Next, we performed ChIP-qPCR experiments with mouse hepatocytes treated with glucagon in the presence of FOXA2 knockdown. While glucagon expectedly increased TET3 association with P2 (but not P1) (Fig. 5a), FOXA2 knockdown (Fig. 5b) abolished this effect (Fig. 5a), demonstrating that FOXA2 is required for glucagon stimulated association of TET3 to the P2 promoter. To test whether FOXA2 physically interacts with TET3, co-IP experiments were carried out using hepatocytes overexpressing TET3. FOXA2 was readily detected in TET3-containing complexes, and vice versa (Fig. 5c), supporting a physical interaction of TET3 and FOXA2. It remains to be determined whether the interaction was direct or indirect via yet unidentified factors. Notably, FOXA2 occupied P2 irrespective of glucagon stimulation; but FOXA2 knockdown diminished FOXA2 association with P2 (Fig. 5d), consistent with previous reports showing FOXA2 binding to its target promoters constitutively regardless of metabolic state[32].

Importantly, co-IP experiments showed increased interaction between FOXA2 and TET3 in response to glucagon stimulation as compared to no glucagon stimulation (Fig. 5e). Based on these data we propose that the constitutive association of FOXA2 with the P2 promoter maintains an open chromatin structure; upon glucagon stimulation TET3 rises above a critical threshold level and is effectively recruited to the P2 promoter via interaction with FOXA2.

As the P2 isoform is specifically induced during fasting, we hypothesized that this isoform is more potent than the P1 isoform in transactivation of gluconeogenic genes. Transcription activation of *PCK1* and *G6PC* by HNF4α requires PGC-1α which by itself does not bind DNA[3,4]. We used a luciferase reporter (gAF1[4]) driven by a fragment of the *PCK1* promoter harboring a HNF4α-binding site. Plasmids expressing human HNF4α8 (representing a P2-derived isoform)[33] or HNF4α2 (representing a P1-derived isoform)[34] were transfected into U-2 OS cells, together with gAF1, a *Renilla* luciferase reporter for transfection normalization[4], and increasing amounts of a PGC-1α expression vector[35]. The transcriptional activity of HNF4α8 was significantly higher than that of HNF4α2 in increasing concentrations of PGC-1α, although HNF4α8 and HNF4α2 were expressed at comparable levels (Supplementary Fig. 5). These results suggest that the P2 isoform, when co-activated by PGC-1α, is likely a more potent transactivator of gluconeogenic genes. It remains to be determined which of the three P2 isoforms (α7–α9) are responsible for gluconeogenic activation in vivo.

**Targeting TET3 or the P2 isoform improves glucose metabolism.** Excessive HGP in T2D is primarily a result of dysregulated gluconeogenesis, a major contributor to impaired glucose homeostasis[36]. Mice with T2D exhibit increased hepatic expression of TET3 and HNF4α P2 isoform (Fig. 2). Human T2D patients showed increased *TET3* expression in the liver

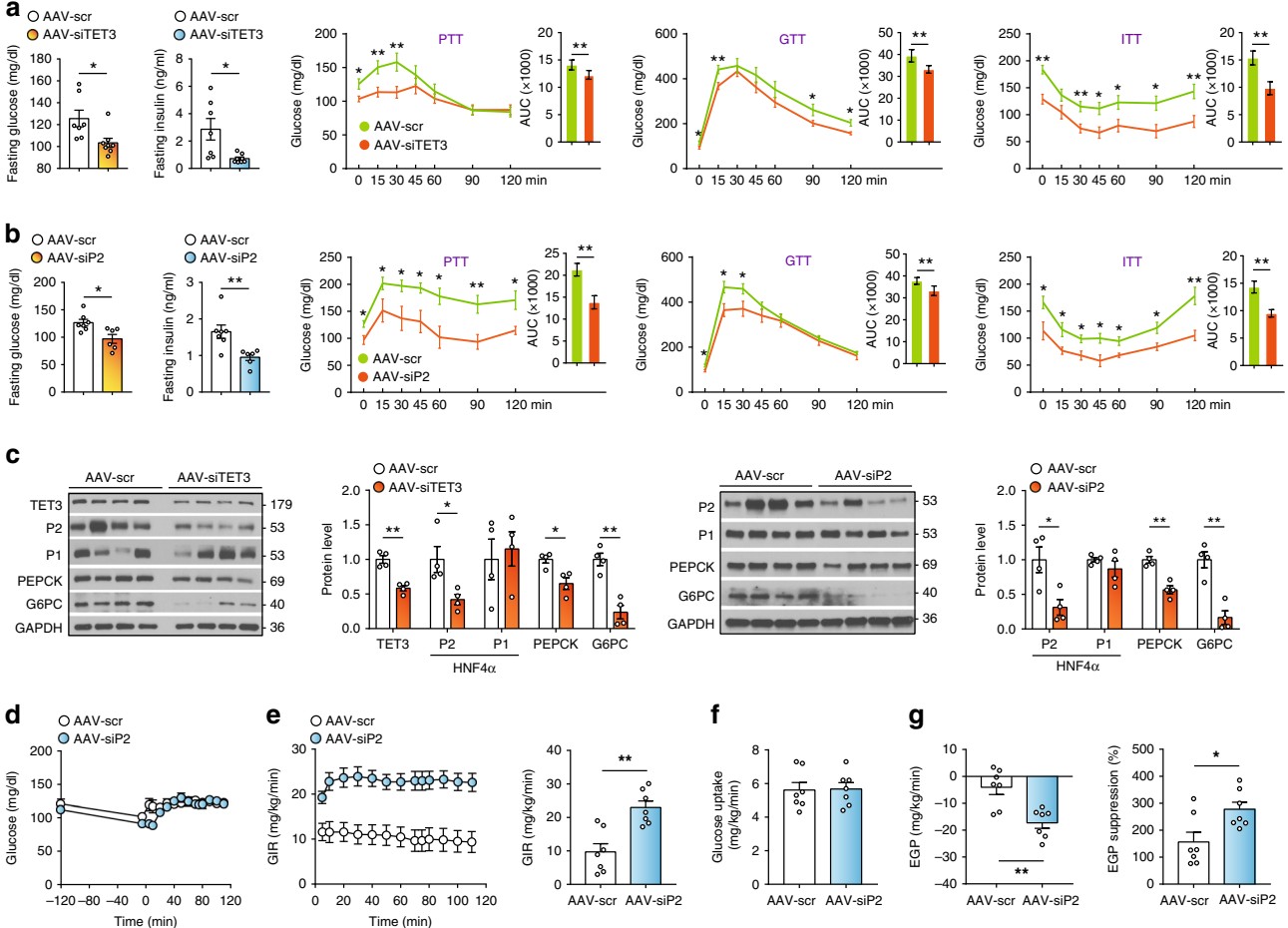

**Fig. 6 The P2 isoform epigenetically induced by TET3 contributes to increased HGP in HFD mice. a** Fasting blood glucose and insulin levels (two-tailed Student's $t$ tests), and PTT, GTT, and ITT in HFD mice infected with AAV-scr or AAV-siTET3 for 10 days. $n = 7–8$. Two-way ANOVA with Sidak post-test. **b** Fasting blood glucose and insulin levels (two-tailed Student's $t$ tests), and PTT, GTT, and ITT in HFD mice infected with AAV-scr or AAV-siP2 for 10 days. $n = 6–7$, Two-way ANOVA with Sidak post-test. **c** IB of TET3, HNF4α P2 and P1 isoforms, and PEPCK and G6PC in liver tissues from HFD mice infected with AAV-scr, AAV-siTET3, or AAV-siP2 for 10 days. $n = 4$. **d–g** Hyperinsulinemic–euglycemic clamp studies from mice fed HFD for 12 weeks and infected with AAV-scr or AAV-siP2 for 10 days. $n = 7$, Two-way ANOVA with Sidak post-test. Data **a–c** are representative of two independent experiments. All data are presented as mean ± SEM. *$p < 0.05$, **$p < 0.01$. AUC area under the curve. Source data are provided as a Source Data File.

(Supplementary Fig. 3c). We asked whether inhibition of TET3 or the P2 isoform would reduce HGP thereby improving glucose homeostasis, using both HFD and genetic ($Lep^{ob/ob}$) mouse models of T2D[4]. We infused the diabetic mice with AAV-scr, AAV-siTET3, or AAV-siP2, and the effects of gene knockdown on glucose metabolism were assessed 10 days later. In the HFD animals, knockdown of TET3 significantly decreased fasting blood glucose, fasting insulin, and PTT (Fig. 6a), suggesting decreased hepatic gluconeogenesis, which was further supported by decreased protein levels of HNF4α P2, PEPCK, and G6PC in the liver (Fig. 6c). Glucose tolerance tests (GTT) and insulin tolerance tests (ITT) showed significantly enhanced glucose tolerance and insulin sensitivity in TET3 knockdown as compared to control animals (Fig. 6a). Effects of HNF4α P2 knockdown mirrored those of TET3 knockdown (Fig. 6b, c). Similar observations were made using $Lep^{ob/ob}$ mice (Supplementary Fig. 6a–d). To further examine the cause of alterations in blood glucose, hyperinsulinemic–euglycemic clamp studies were conducted in HFD mice infected with AAV-siP2. Compared to the AAV-scr-treated mice, the AAV-siP2-treated mice showed significantly higher glucose infusion rate (GIR) to maintain euglycemia (Fig. 6d, e), reflecting increased insulin sensitivity. This was not due to increased peripheral glucose uptake (Fig. 6f) but rather

to increased insulin-stimulated suppression of endogenous glucose production (EGP) (Fig. 6g). These results further support the notion that inhibiting expression of the P2 isoform improves hyperglycemia and insulin sensitivity via reduction in HGP. While in the adult pancreas both P2 and P1 promoters of HNF4α were reported to be active and important for pancreas β-cell function[37], the unchanged expression of P2 and P1 isoforms in mice infected with AAV-siP2 (Supplementary Fig. 6e) ruled out the possibility that the decreased fasting blood insulin was caused by altered HNF4α in the pancreas.

## Discussion

As illustrated in Fig. 7, here we report HNF4α P2 promoter reactivation in adult liver and its critical role in control of HGP under both physiological and pathological conditions. The regulation involves an epigenetic mechanism mediated by a member of the TET family proteins not previously known to have a role in glucose homeostasis. Importantly, this TET3-mediated P2 promoter reactivation in HGP appears to be conserved between human and mouse. It is noteworthy that the preferential enrichment of FOXA2 at the P2 versus the P1 promoters as demonstrated by ChIP from both human (Supplementary Fig. 4) and mouse (Fig. 4i) livers does not exclude the possibility of

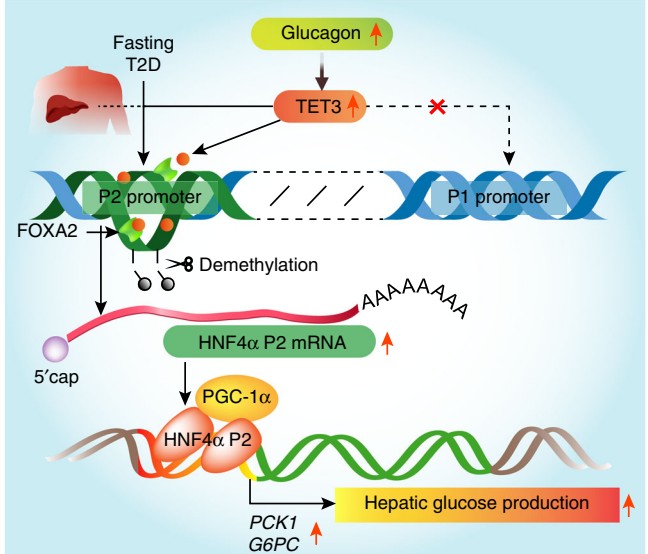

**Fig. 7** A proposed model. Constitutive association of FOXA2 with the P2 promoter of *HNF4α* maintains an open chromatin structure. During fasting (physiological) or under chronic diabetic conditions (pathological), glucagon upregulates TET3 above a critical threshold level and/or allows it to be effectively recruited to the P2 promoter via interaction with FOXA2. Binding of TET3 induces 5hmC deposition and subsequent demethylation of the P2 promoter, leading to increased transcription and production of the P2-specific isoform contributing to gluconeogenesis activation.

FOXA2 binding to the P1 promoter, albeit to a lesser extent. As both FOXA2 and TETs have been shown to act at numerous genomic loci as assessed by ChIP-seq and genome-wide 5hmC profiling[9,38], it is plausible that interactions between FOXA2 and TET3 (directly or indirectly via other factors) may facilitate expression of other genes involved in gluconeogenesis and/or other metabolic processes, and this warrants future investigation. Finally, we show that inhibition of TET3 or only the P2-specific isoform alleviates T2D in mouse models. Our findings may provide a mechanistic link between the P2 promoter SNPs and increased risk of T2D[39,40] and suggest that targeting TET3, the P2-specific isoform, or both, have therapeutic potential for T2D.

## Methods

**Animals**. All animal work was approved by the Yale University Institutional Animal Care and Use Committee and was conducted accordingly. All mice used in this report were male. Mice were housed at 22–24 ºC with a 12 h light/12 h dark cycle with normal chow (NC) (Harlan Teklad no. 2018, 18% calories from fat) or high-fat diets (HFD) (Research Diets, D12451, 45% calories from fat) and water provided ad libitum. The H19 KO and WT mice on a background of C57BL/6J have been previously described[17,18]. To induce insulin resistance (Fig. 2c, d), WT C57BL/6J mice were exposed to HFD for 11 days or 12 weeks as previously described[18]. The diet-induced obese and diabetic mice (19–20-week old, fed HFD starting at age of 6 weeks) and *Lep^{ob/ob}* mice (8–9-week old) used in experiments were purchased from the Jackson Laboratories. Before experiments, mice were allowed to acclimate for at least 7 days in our animal facility.

**PTT, GTT, and ITT**. PTT and GTT were performed following 16 and 12 h overnight fasting, respectively. Each animal received an i.p. injection of 2 g/kg pyruvate (Sigma-Aldrich, cat#P5280) or 2 g/kg of glucose (Sigma-Aldrich, G5767) in sterile saline. ITT were performed following morning fasting (3 h). Each animal received an i.p. injection of 1 U/kg insulin (Humulin R; Eli Lilly). Blood glucose concentrations were measured using Breeze2 glucometer (Bayer) via tail vein bleeding at the indicated time points after injection. Plasma insulin levels were measured using Mouse Insulin ELISA kit (EMD Millipore, Billerica, MA, catalog # EZRMI-13K) according to the manufacturer's instructions. For all experiments, age-matched animals were used. For RNA and protein analyses, tissues were collected following euthanasia, snap frozen in liquid nitrogen, and stored at −80 °C until use. For information on animal numbers, refer to figure legends.

**Hyperinsulinemic–euglycemic clamp studies**. The hyperinsulinemic–euglycemic-clamp was conducted using previous methods[41]. In brief, mice fed the HFD for 12 weeks were catheterized in jugular veins with polyethylene catheters under deep anesthesia. Mice were singly housed for 3–5 days recovery period after surgery. The hyperinsulinemic–euglycemic-clamp experiments were performed under conscious and unrestrained mice after 16 h overnight fasting. The protocol consisted of a 120 min basal period ($t = -120–0$ min) followed by a 115 min clamp period ($t = 0–115$ min). [3-$^3$H] glucose (5 µCi; Perkin Elmer) was given at $t = -120$ min followed by a 0.05 µCi/ min infusion for 2 h. During the basal period, at $t = -15$ and $-5$ min, blood samples were taken for the assessment of basal glucose level and glucose turnover. The clamp period was begun at $t = 0$ min with primed and continuous infusion of human insulin (8 mU/kg bolus followed by a rate of 2.5 mU kg$^{-1}$ min$^{-1}$; Humulin R; Eli Lilly). Blood glucose was measured by glucometer (Breeze 2; Bayer HealthCare LLC) at 10 min intervals, and 30% glucose was infused at a variable rate in order to maintain euglycemia (110–130 mg/dl). Blood samples were collected every 10 min from $t = 70$ to 115 min and processed to determine glucose-specific activity.

**Virus production and in vivo virus administration**. The Ad-GFP adenovirus was purchased (CV10001, Vigene Biosciences Inc.). The Ad-TET3 virus was custom made by cloning a mouse full-length TET3 ORF PCR amplified from pcDNA-Flag-Tet3[42] (60940, Addgene) into an Ad-TBG vector that drives TET3 expression from a liver-specific thyroxine-binding globulin (TBG) promoter (Vigene Biosciences, Rockville, MD 20850, USA). The AAV-H19 virus was previously described[17]. The negative control siRNA virus AAV-scr was purchased (iAAV01500, Applied Biological Materials Inc.). The AAV-siTET3 (iAAV04929608), AAV-sihTET3 (iAAV02441008), AAV-sihP2, AAV-siFOXA2 (iAAV04346808) viruses were purchased from Applied Biological Materials Inc. AAV-siP2 and AAV-sihP2 were custom made by Applied Biological Materials Inc. The targeted sequences for AAV-siP2 and AAV-sihP2 are 5′-CCTTTGCTGCTGTGTGTGTGGGCCCCTGCTC and 5′-ATGGTCAGCGTGAACGCGCCCTCGGGGC, respectively. All AAV viruses were serotype 8. Mice were tail vein injected with indicated viruses at $1 \times 10^{10}$ gc/mouse (adeno) or $2 \times 10^{10}$ gc/mouse (AAV), in 150 µl of PBS/0.05% sorbitol. Before injection, mice were exposed to heat lamp to dilate the tail vein and then placed in a restrainer permitting access to the tail vein. The tail was cleansed with 70% ethanol and the injection was made in the lateral vein, using 30-gauge needles.

**Primary hepatocytes and viral infection**. Mouse primary hepatocytes were isolated by the Yale Liver Center according to previous protocols[17]. Cells were maintained in complete CM (Williams' Medium [GIBCO,12551] supplemented with 5% FBS, 1% 1 M HEPES buffer [GIBCO, 15630-080], 1% L-glutamine [GIBCO, 25030-081], 1% SPA [GIBCO, 15240-062], 4 mg/l insulin [GIBCO, 12585-014], and 1 µM dexamethasone [Sigma, D4902]). For TET3 overexpression experiments, freshly isolated mouse primary hepatocytes were seeded in 12-well collagen 1-coated plates (BD Biosciences Discovery Labware) at $4 \times 10^5$ cells/well and infected with Ad-GFP or Ad-TET3 at 4000 gc/cell at 3 h after seeding. For TET3 or HNF4α P2 isoform knockdown experiments, cells seeded in 12-well plates were infected with AAV-scr, AAV-siTET3, or AAV-siP2 at 6000 gc/cell at 3 h after seeding. Medium was changed the next day and glucagon was added at a final concentration of 20 nM in CM. RNA and protein were isolated and glucose production assays were carried out at the time points indicated in the figure legends.

Human primary hepatocytes (M00995, Millipore, Sigma) were purchased, thawed, and maintained in InVitroGRO™ CP Medium (Sigma-Aldrich, Z990003) containing 2% Torpedo™ Antibiotic Mix. The cells were seeded in 12-well plates coated with collagen 1 (Millipore, 08-115). Infection of AAV-scr, AAV-sihTET3, and AAV-sihP2 were performed the next day after cell thawing and followed the same dosing protocol for the mouse primary hepatocytes.

**Luciferase reporter assays**. Human U-2 OS cell line (92022711) was purchased as authenticated from Sigma Aldrich and maintained in McCoy's 5a medium supplemented with 1.5 mM glutamine and 10% FBS. The assays were carried out in a 48-well plate scale as previously described[43] with minor modifications. Transfections were performed with lipofectamine 3000 (Thermo Fisher Scientific) with a fixed total quantity of DNA (300 ng per well). 100 ng of gAF1 (*Pck1* firefly luciferase reporter) and 0.2 ng of *Renilla* luciferase construct (for transfection efficiency normalization)[4], together with 0, 10, 20, 40, or 80 ng of pAd-Track Flag HA PGC-1 alpha (expresses PGC-1α, #14426, Addgene)[35] and 25 ng of FR_HNF4A8 (expresses HNF4α8, #31114, Addgene)[33] or FR_HNF4A2 (expresses HNF4α2, #31100, Addgene)[34] were transfected into U-2 OS cells. To prepare transfection solution for each well, plasmids were mixed with 25 µl OPTI-MEM and 1 µl P3000 by gentle pipetting. In parallel, 0.75 µl Lipofectamine 3000 was mixed with 25 µl OPTI-MEM. Following 5 min of incubation at room temperature (RT), the two were mixed by gentle pipetting and incubated for 10 min at RT to allow plasmids/lipid complexes to form. At the end of incubation, the 50 µl transfection solution was used to re-suspend cell pellet ($4 \times 10^4$ cells). After incubation at RT for 10 min, regular growth medium was added at a ratio of 1:5 (1 volume of transfection solution/5 volumes of growth medium) and the cell suspension was transferred to one well of a 48-well plates. Experiments were run in triplicate. Luciferase activities were measured 24 h post-transfection using Promega Dual-Luciferase Reporter

Assay System (E1960) according to the manufacturer's protocol. Data are presented as firefly luciferase reporter values normalized to *Renilla* values.

**Quantitative methylation-specific PCR**. Genomic DNA was extracted from mouse primary hepatocytes grown in 12-well plates using Quick-gDNA MicroPrep (Zymo Research Corporation, D3021) according to the manufacturer's instructions. For bisulfite treatment, 480 ng of DNA was used for each column using EZ DNA Methylation-Gold Kit (Zymo, D5006). 100 μl of elution buffer was used to elute DNA from each column. Real-time quantitative PCR was performed in a 15 μl reaction containing 5 μl of the eluted DNA using iQSYBRGreen in a Bio-Rad iCycler. Two sets of PCR primers were designed: one for unmethylated and one for methylated DNA sequences. The PCR primers for methylated DNA were used at a final concentration of 0.6 μM in each PCR reaction. PCR was performed by initial denaturation at 95 °C for 5 min, followed by 40 cycles of 30 s at 95 °C, 30 s at 60 °C, and 30 s at 72 °C. Specificity was verified by melting curve analysis. The threshold cycle (Ct) values of each sample were used in the post-PCR data analysis. The ratio of methylated versus unmethylated DNA sequences are presented. The primers used for QMSP are listed in Supplementary Table 1.

**Chromatin immunoprecipitation-quantitative PCR**. Experiments were performed in a 12-well plate scale using the Pierce Agarose ChIP Kit (Thermo Scientific, 26156) according to the manufacturer's instructions with minor modifications. Briefly, agarose beads were used to pre-bind with antibodies (anti-TET3, anti-Ser-5(P)-RNAP, anti-FOXA2, or pre-immune IgGs as negative controls) overnight at 4 °C. The next day, hepatocytes were crosslinked with 1% formaldehyde at RT for 10 min, and the reaction was stopped by 1x glycine. ChIPs were carried out overnight at 4 °C. Levels of ChIP-purified DNA were determined with qPCR (see Supplementary Table 1 for primer sequences). The relative enrichments of the indicated DNA regions were calculated using the Percent Input Method according to the manufacturer's instructions and are presented as % input.

**Hydroxymethylated DNA immunoprecipitation coupled with qPCR**. The experiments were carried out using the EpiQuik hMeDIP Kit (P-1038-48, Epigentek) according to the manufacturer's instructions. Briefly, mouse primary hepatocytes maintained in 12-well plates at $4 \times 10^5$ cells/well were stimulated with 20 nM of glucagon (vehicle as control) or infected with Ad-TET3 (Ad-GFP as control) at 4000 gc/cell at 3 h after seeding. gDNAs were extracted using Quick gDNA MicroPrep (D3021, Zymo Research Corporation) at 24 h (glucagon) or 36 h (adeno) later. gDNAs were sheared using a Qsonica Sonicators Q125 probe, with a setting of nine pulses of 10 s each at 35% amplitude followed by a 40 s rest period on ice between each pulse. Sheared DNA fragments (ranged in size from 200 to 600 bps as assessed by agarose gel electrophoresis) were immunoprecipitated using the 5hmC rabbit polyclonal antibody from the kit. qPCR was performed in a 25 μl reaction containing 2.5 μl of the eluted DNA using iTAC SYBRGreen in a Bio-Rad iCycler. The Ct values of each sample were used in the post-PCR data analysis. The relative enrichments (after normalization against control IgG) of the indicated DNA regions were calculated using the Percent Input Method according to the manufacturer's instructions.

**Glucose production assay**. Glucose production assays were performed using Amplex Red Glucose/Glucose Oxidase Assay Kit (Molecular Probes, Invitrogen, A22189), according to the manufacturer's instructions. Briefly, primary hepatocytes grown in 12-well plates were used. On the day of the assay, CM was replaced with glucose-free and phenol red-free DMEM (Gibco, A14430-01) supplemented with 2 mM L-glutamine and 15 mM HEPES for 2 h. Then, cells were incubated in 120 μl of glucose production medium (glucose-free and phenol red-free DMEM, 20 mM sodium lactate, 2 mM sodium pyruvate, and 0.5% BSA, 2 mM L-glutamine, and 15 mM HEPES) for 4 h. Supernatants (50 μl) were used for measurements of glucose concentration, which was normalized to total protein content of cells.

**RNA extraction and RT-qPCR**. Total RNAs were extracted from liver or pancreas tissues or from primary hepatocytes using PureLink RNA Mini Kit. cDNA was synthesized using PrimeScript RT Reagent Kit in a 20 μl reaction containing 0.5–1 μg of total RNA. Real-time quantitative PCR was performed in a 15 μl reaction containing 0.5–1 μl of cDNA using iQSYBRGreen in a Bio-Rad iCycler. PCR was performed by initial denaturation at 95 °C for 5 min, followed by 40 cycles of 30 s at 95 °C, 30 s at 60 °C, and 30 s at 72 °C. Specificity was verified by melting curve analysis. The Ct values of each sample were used in the post-PCR data analysis. Gene expression levels were normalized against house-keeping genes GAPDH and RPLP0. Real-time PCR primers are listed in Supplementary Table 1.

**Immunoblot analysis**. Primary hepatocytes in 12-well plates were dissociated with 0.25% trypsin. After centrifugation at $500 \times g$ for 5 min, supernatants were discarded and cell pallets were homogenized in 2x SDS-sample buffer (100 μl/well), followed by heating at 100 °C for 5 min, with occasional vortexing. For liver tissue samples, 5 mg of tissues were minced and 200 μl of 2xSDS-sample buffer was added, followed by homogenization on ice using a sonicator (Qsonica, Q125-110). Homogenized samples were heated at 100 °C for 5 min and then centrifuged at 12,000×g for 5 min to remove insoluble materials before loading onto 12% SDS gels

(5 μl/well), followed by Western blot analysis. Bands on Western blot gels were quantified using ImageJ. GAPDH was used as a loading control. Antibodies used in Western blot analysis with dilutions are presented in Supplementary Table 2.

**Immunoprecipitation**. To prepare antibodies, 8 μl (packed volume) of ChIP grade Protein A/G beads (Pierce Agarose ChIP Kit, Thermo Scientific, 26156) were washed twice with 0.5 ml of IP buffer (1% Triton X-100, 300 mM NaCl, 10 mM Tris–HCl at pH 7.5, and 10 mM EDTA), followed by incubation with 5 μg of rabbit polyclonal anti-TET3 (Millipore Sigma, ABE290), goat polyclonal anti-FOXA2 (R&D systems, AF2400) or preimmune rabbit or goat IgG in 200 μl of IP buffer at 4 °C overnight. Antibody bound beads were pelleted by brief centrifugation and kept on ice until use. To prepare cell lysate, H19 KO mouse hepatocytes ($1 \times 10^6$ cells per IP) infected with Ad-TET3 for 24 h (or WT mouse hepatocytes stimulated with glucagon for 24 h) were washed on plate once with ice-cold PBS, followed by incubation on ice for 10 min in 400 μl of freshly prepared gentle lysis buffer (GLB: 1% Triton X-100, 10 mM NaCl, 10 mM Tris–HCl at pH 7.5, 10 mM EDTA, 1 mM PMSF, 1 mM DTT, and 1x protease inhibitor cocktail [Calbiochem]). The cells together with the lysis buffer were manually scraped into a 1.5 ml tube, followed by incubation on ice for an additional 15 min. After centrifugation at 12,000 × g at 4 °C for 15 min to remove insoluble materials, the lysate was transferred to a new tube with addition of 5 M NaCl to a final concentration of 150 mM. The lysate was pre-cleared with 8 μl of washed Protein A/G beads by rotating at 4 °C for 1 h to minimize non-specific binding, followed by centrifugation at 12,000 × g at 4 °C for 15 min to remove insoluble materials. The cleared lysate was transferred to a new tube containing antibody/preimmune IgG-coated beads (350 μl of lysate per IP). IP was carried out at 4 °C overnight. Following IP, beads were quickly washed twice with 1 ml of cold IP buffer and then washed additional three times with IP buffer (300 mM salt) by rotating at 4 °C for 3 min each time. After the final wash, residual liquid was completely removed and the beads were eluted with 20 μl of 2xSDS sample buffer at 100 °C for 3 min. Five μl per gel well of eluant was loaded onto 10% SDS–PAGE gradient gels (Bio-rad). For Western analysis, anti-TET3 and anti-FOXA2 were diluted at 1:200 and 1:500, respectively. The secondary antibodies used were Rabbit IgG TrueBlot® and Goat IgG True-Blot®, respectively. These unique HRP-conjugated monoclonal secondary antibodies enable detection of immunoblotted target proteins without hindrance by interfering immunoprecipitating immunoglobulin heavy and light chains.

**Cell viability analysis**. These were performed using the CellTiter-Glo 2.0 Assay (G9242, Promega) according to the manufacturer's protocol. Results are presented as percentage of viabilities of freshly isolated cells which were arbitrarily set as 100%.

**Statistical analysis**. The number of independent experiments and the statistical analysis for each figure are indicated in the legends. All statistical analyses were performed using GraphPad Prism version 7.01 for Windows (GraphPad Software, La Jolla, CA, USA, www.graphpad.com) and are presented as mean ± SEM. Two-tailed Student's *t* tests (or as otherwise indicated) were used to compare means between groups. $p < 0.05$ was considered significant. For integrate analysis of human liver TET3 expression datasets, the Robust Rank Aggregation (RRA) method[44] was used.

**Reporting summary**. Further information on research design is available in the Nature Research Reporting Summary linked to this article.

## Data availability

The data that support the findings of this study are available from the corresponding author upon request. The source data underlying Figs. 1b, e, h, l; 2a–d, i–l, m, o, p, r; 3h; 5b, c, e; 6c; S3d; S5b and S6d are provided as Source Data File.

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

## Acknowledgements

We thank Kathy Harry at the Yale Liver Center for isolating primary mouse hepatocytes, Drs. Yi Zhang for pcDNA-Flag-Tet3, Gerhart Ryffel for FR_HNF4A2, and FR_HNF4A8, and Pere Puigserver for pAd-Track Flag HA PGC-1 alpha, gAF1, and *Renilla* plasmid constructs. Research reported in this publication was supported by grants from the National Institute of Diabetes and Digestive and Kidney Diseases of the National Institutes of Health (DK119386 to Y.H., DK097566, DK107293, and DK120321 to S.D., and DK089098 to X.Y.), and grant GE001347 from the Mckern Albert foundation to Y.H.

## Author contribution

D.L., T.C., and X.S. performed most of the experiments. D.X. and X.H. performed 5hMeDIP and co-IP experiments. S.J. and S.D. designed and performed the hyperinsulinemic/euglycemic clamp studies and analyzed and interpreted the data. Y.H. initiated, designed, and led the study, and D.L. and Y.H. wrote the manuscript. S.D., X.Y., G.G.C. and H.S.T. provided intellectual input and critical reading of the manuscript.

## Competing interests

The authors declare no competing interests.
