## [Peer Review File · Nature Communications]

Reviewers' comments:

Reviewer #1 (Remarks to the Author):

Li et al in their manuscript entitled "Hepatic TET3 contributes to type-2 diabetes by inducing the HNF4 α fetal isoform" reports a novel role of TET3, which is a DNA demethylase, in regulation of hepatic glucose output via upregulation of HNF4 α fetal isoform (P2 isoform).

The P2 isoform of HNF4 α was thought to be involved in fetal liver and did not have significant functions in adult livers. Authors document that this isoform, contrary to general expectations, is induced by glucagon and fasting through a mechanism involving TET3. They show that TET3 leads to hypomethylation of specifically P2 promoter and that this increase in P2 levels regulates hepatic glucose output in primary hepatocytes. Author also document that this pathway is involved in vivo and by performing PTT tests and knocking down P2 isoform authors show that P2 isoform regulates gluconeogenesis in vivo. Furthermore, when P2 isoform of TET3 is depleted in the livers of obese and diabetic mouse models, author document a significant improvement in glucose homeostasis.

The findings in this paper is novel and interesting and should be published without delay.

Reviewer #2 (Remarks to the Author):

This is a revised paper that determine the mechanism underlying the regulation of hepatic glucose production, focusing on the regulation of two alternative promoters in the liver-enriched transcription factor Hnf4a gene (P1 and P2) that are differentially expression in adult and fetal, respectively. The major finding reported here is that glucagon stimulates the expression of Tet3 DNA dioxygenase, which then binds to P2 and stimulates the expression from this promoter. Several

major issues and a number of technical issues were raised in the previous round of review. The authors addressed most of them, but two remain.

One major issue I previously raised was how Tet3 is recruited to P2 upon glucagon stimulation. In the revised paper, the authors provided new results showing that transcription factor FOXA2 binds to P2, but not P1 and interacts with Tet3 and, when knocked down, reduced the binding of Tet3 to P2. These are good addition, but newly added results missed two critical questions of this paper: (1) How does glucagon stimulate Tet3 to bind P2 if “FOXA2 occupied P2 (but not P1) irrespective of glucagon stimulation”? Is it simply due the increased expression of Tet3 after glucagon stimulation? If so, can authors demonstrate this by expanding the Extended Fig. 5d to include +/- glucagon treatment? (2) How specific does FOXA2 selectively recruit Tet3 to P2? Author identified FOXA2 binding/enrichment on P2 through inspection of previous ENCODE ChIP-seq data set, but did not provide any information on how many genes, especially those regulated by glucagon, are also bound by FOXA2. If FOXA2-Tet3 complex is assembled simply by the increased Tet3 expression after glucagon, how many genes are regulated by FOXA2-Tet3? A standard Tet3 ChIP-seq of in cells expressing and knockdown FOXA2 would be very informative.

I also suggested authors to include 5hmC and 5mC ChIP of HNF4 α upon TET3 overexpression and glucagon treatment. Authors responded by that their ‘focus in the current study is the endpoint of TET3 action (DNA demethylation) and how it affects RNA polymerase binding and transcription’, and the Tet3 binding results ‘support our hypothesis that TET3 promotes P2--specific transcription by binding and demethylating it’. And whether TET3--induced 5hmC itself plays a role in P2 reactivation warrants further investigations that are beyond the scope of the current manuscript”. I disagree with these arguments. 5hmC ChIP has become the standard assay for demonstrating the direct binding and function of a Tet enzyme on a given gene. The main purpose of including 5hmC ChIP is not to determine how it affects the binding of which factor. Rather, it provides a specific assay to validate the Tet2 binding and action since only known enzyme producing 5hmC is Tet and 5hmC antibody quality is more consistent with Tet antibody, especially for ChIP.

Reviewer #3 (Remarks to the Author):

In this manuscript, the authors determined the regulatory effect of TET3 on the expression of the P2 isoform of Hnf4 α , which is predominantly expressed in the fetal liver. They found that TET3 promoted hepatic glucose production (HGP), and played an important role in the upregulated HGP observed in type 2 diabetes animal models such as HFD and ob/ob mice. The authors suggested that TET3 bound specifically to the P2 promoter via its interaction with FOXA2. The increased P2 HNF4 α

proteins would then increase the gluconeogenic genes and promote glucose output. While the authors perform quite all the experiment suggested by the reviewers and add additional information on the biological role of TET3 and P2 isoforms on glucose output, there are still some questions raised regarding the in-vitro experiments and particularly the specific role of glucagon and FOXA 2 on the P2 promoter regulation.

1. In the experiments performed in primary hepatocytes, cells were maintained in a medium containing serum, insulin and dexamethasone. The stimulatory effect of glucagon was studied in addition to the other hormones (and in addition to the glucagon that might be present in the serum). P1 promoter is known to be inhibited by insulin. To assess that glucagon specifically target the P2 promoter but not the P1, the authors should study the effect of glucagon in the absence of insulin. Indeed, in the presence of insulin, some transcription factors might be present on the P1 promoter and avoid a putative effect of glucagon on this region. This might be discussed in the results.

2. The duration of the AAV and glucagon treatment raised concerns regarding hepatocytes viability (which was not given). Glucose output should be expressed as absolute data and not as data relative to the control situation.

3. FOXA 2 is known to bind to P1 promoter. Why the authors did not reproduce this published result in the Extended Data Figure 5 ?

A. Gautier-Stein

Point-by-point response to referees' comments

Reviewer #1

Li et al in their manuscript entitled "Hepatic TET3 contributes to type-2 diabetes by inducing 1 the HNF4 α fetal isoform" reports a novel role of TET3, which is a DNA demethylase, in regulation of hepatic glucose output via upregulation of HNF4 α fetal isoform (P2 isoform).

The P2 isoform of HNF4 α was thought to be involved in fetal liver and did not have significant functions in adult livers. Authors document that this isoform, contrary to general expectations, is induced by glucagon and fasting through a mechanism involving TET3. They show that TET 3 leads to hypomethylation of specifically P2 promoter and that this increase in P2 levels regulates hepatic glucose output in primary hepatocytes. Author also document that this pathway is involved in vivo and by performing PTT tests and knocking down P2 isoform authors show that P2 isoform is regulates gluconeogenesis in vivo. Furthermore, when P2 isoform of TET3 is depleted in the livers of obese and diabetic mouse models, author document a significant improvement in glucose homeostasis.

The findings in this paper is novel and interesting and should be published without delay.

Response: Thank you very much!

Reviewer #2

This is a revised paper that determine the mechanism underlying the regulation of hepatic glucose production, focusing on the regulation of two alternative promoters in the liver-enriched transcription factor Hnf4a gene (P1 and P2) that are differentially expression in adult and fetal, respectively. The major finding reported here is that glucagon stimulates the expression of Tet3 DNA dioxygenase, which then binds to P2 and stimulates the expression from this promoter. Several major issues and a number of technical issues were raised in the previous round of review. The authors addressed most of them, but two remain.

1. One major issue I previously raised was how Tet3 is recruited to P2 upon glucagon stimulation. In the revised paper, the authors provided new results showing that transcription factor FOXA2 binds to P2, but not P1 and interacts with Tet3 and, when knocked down, reduced the binding of Tet3 to P2. These are good addition, but newly added results missed two critical questions of this paper: (1) How does glucagon stimulate Tet3 to bind P2 if "FOXA2 occupied P2 (but not P1) irrespective of glucagon stimulation"? Is it simply due the increased expression of Tet3 after glucagon stimulation? If so, can authors demonstrate this by expanding the Extended Fig. 5d to include +/- glucagon treatment?

Response 1: We appreciate the reviewer's insightful comments and suggestions. To address these important questions we have performed additional experiments and have also extensively revised the text to be more explicit and clear about our rationale for choosing FOXA2 and how our new data further strengthen our conclusion on the mechanism by which TET3 is recruited to the P2 promoter. Specifically, in the revised manuscript we have added red highlighted paragraphs in the Introduction and Results to suggest that the TET3-FOXA2 mechanism may be similar to the published combination of TET2/TET3-PU.1 (lines 62-69 on pages 3-4) and FOXA2-CREB/GR (lines 201-233 on pages 8-10). Importantly, our new co-IP results (Fig. 4j) show a significantly increased interaction of TET3 with FOXA2 in the presence of glucagon stimulation. We thus propose (lines 230-233 on page 10) that the constitutive association of FOXA2 with the P2 promoter maintains an open chromatin structure; upon glucagon stimulation TET3 is effectively recruited to the P2 promoter via interaction with FOXA2, perhaps by rising above a critical threshold level. Our new Fig. 6 legend also summarizes our main conclusion: Constitutive association of FOXA2 with the P2 promoter of *HNF4 α* maintains an open chromatin structure. During fasting (physiological) or under chronic diabetic conditions (pathological), glucagon upregulates TET3 above a critical threshold level and/or allows it to be effectively recruited to the P2 promoter via interaction with FOXA2. Binding of TET3 induces 5hmC deposition and

subsequent demethylation of the P2 promoter, leading to increased transcription and production of the P2-specific isoform contributing to gluconeogenesis activation.

2. (2) How specific does FOXA2 selectively recruit Tet3 to P2? Author identified FOXA2 binding/enrichment on P2 through inspection of previous ENCODE ChIP-seq data set, but did not provide any information on how many genes, especially those regulated by glucagon, are also bound by FOXA2. If FOXA2-Tet3 complex is assembled simply by the increased Tet3 expression after glucagon, how many genes are regulated by FOXA2-Tet3? A standard Tet3 ChIP-seq of in cells expressing and knockdown FOXA2 would be very informative.

Response 2: Using ChIP-seq FOXA2 has been reported to bind to 45% of expressed genes in the adult mouse liver, acting either as an inducer or repressor of gene expression depending on its binding location (lines 202-204 on pages 8-9 in the revised manuscript). Similarly, genome-wide studies have shown that the TET proteins act at numerous genomic loci. We therefore propose that interactions between FOXA2 and TET3 may facilitate expression of other genes involved in gluconeogenesis and/or other metabolic processes (lines 278-282 on page 11). As the key point of this manuscript is P2 versus P1, we hope that the reviewer will agree we have provided strong experimental data to support our conclusion. The suggested experiment of TET3 ChIP-seq with FOXA2 knockdown certainly has value but will likely distract readers from the main point of our work and also is beyond the scope of our current study.

3. I also suggested authors to include 5hmC and 5mC ChIP of HNF4 α upon TET3 overexpression and glucagon treatment. Authors responded by that their 'focus in the current study is the endpoint of TET3 action (DNA demethylation) and how it affects RNA polymerase binding and transcription', and the Tet3 binding results 'support our hypothesis that TET3 promotes P2-specific transcription by binding and demethylating it'. And whether TET3-induced 5hmC itself plays a role in P2 reactivation warrants further investigations that are beyond the scope of the current manuscript". I disagree with these arguments. 5hmC ChIP has become the standard assay for demonstrating the direct binding and function of a Tet enzyme on a given gene. The main purpose of including 5hmC ChIP is not to determine how it affects the binding of which factor. Rather, it provides a specific assay to validate the Tet2 binding and action since only known enzyme producing 5hmC is Tet and 5hmC antibody quality is more consistent with Tet antibody, especially for ChIP.

Response 3: We agree with the reviewer on these points. Results from our new 5hmC ChIP-qPCR (hMeDIP-qPCR) results using 5hmC-specific antibody further strengthen our conclusion that TET3 directly binds to the P2 promoter inducing 5hmC deposition and subsequent demethylation. Please refer to new Fig. 4c and lines 183-188 on page 8 and lines 397-409 on page 16 in the revised manuscript.

Reviewer #3:

In this manuscript, the authors determined the regulatory effect of TET3 on the expression of the P2 isoform of Hnf4 α , which is predominantly expressed in the fetal liver. They found that TET3 promoted hepatic glucose production (HGP), and played an important role in the upregulated HGP observed in type 2 diabetes animal models such as HFD and ob/ob mice. The authors suggested that TET3 bound specifically to the P2 promoter via its interaction with FOXA2. The increased P2 HNF4 α proteins would then increase the gluconeogenic genes and promote glucose output. While the authors perform quite all the experiment suggested by the reviewers and add additional information on the biological role of TET3 and P2 isoforms on glucose output, there are still some questions raised regarding the in-vitro experiments and particularly the specific role of glucagon and FOXA 2 on the P2 promoter regulation.

1. In the experiments performed in primary hepatocytes, cells were maintained in a medium containing serum, insulin and dexamethasone. The stimulatory effect of glucagon was studied in addition to the other hormones (and in addition to the glucagon that might be present in the serum). P1 promoter is known to be inhibited by insulin. To assess that glucagon specifically target the P2 promoter but not the P1, the authors should study the effect of glucagon in the absence of insulin. Indeed, in the presence of insulin, some transcription factors

might be present on the P1 promoter and avoid a putative effect of glucagon on this region. This might be discussed in the results.

Response 1: In light of these comments we have added discussion of our rationale for performing glucagon stimulation in the presence of insulin (lines 100-106 on page 5): As stated in our Methods section, all primary hepatocyte experiments (e.g., glucagon stimulation and TET3 overexpression) were performed on cells maintained in a complete culture medium containing serum, insulin and dexamethasone, conditions optimized and important for cell viability. These conditions allowed cell viability to persist to the end of the experiments (Supplementary Figure 3b). Our additional rationale for performing glucagon stimulation in the presence of insulin was derived from the fact that insulin is present in the circulation during fasting, albeit at a lower level as compared to fed conditions. Most importantly, our *in vitro* results derived from primary hepatocytes strongly corroborate our *in vivo* results in supporting our main conclusion.

2. The duration of the AAV and glucagon treatment raised concerns regarding hepatocytes viability (which was not given). Glucose output should be expressed as absolute data and not as data relative to the control situation.

Response 2: We have added the cell viability results to the revised manuscript (Supplementary Fig. 3b, and lines 463-465 on page 18) showing that the cell viability persisted at the end of experiments. We have also revised the figures to show absolute data in new Figs. 1c, 1f, 3e, and 3k.

3. FOXA 2 is known to bind to P1 promoter. Why the authors did not reproduce this published result in the Extended Data Figure 5?

Response 3: Thanks much for being accurate. In light of this comment we have added discussion on page 11 (lines 276-278) in the revised manuscript: It is noteworthy that the preferential enrichment of FOXA2 at the P2 versus the P1 promoters as demonstrated by ChIP from both human (Supplementary Figure 4) and mouse (Fig. 4i) livers does not exclude the possibility of FOXA2 binding to the P1 promoter, albeit to a lesser extent.

REVIEWERS' COMMENTS:

Reviewer #2 (Remarks to the Author):

This is a re-revised paper aimed at determining the mechanism underlying the regulation of hepatic glucose production. The main finding is that glucagon stimulates the expression of Tet3 DNA dioxygenase, which then binds to P2 and stimulates the expression from this promoter. In the comments to the last revision, I raised three issues. The authors provide new experiments to address two of them. The data support the conclusion that glucagon stimulates FOXA2-Tet3 association (new figure 4j) and 5hmC on P2 but not P1 (new Figure 4c). I am satisfied with these two new results.

The remaining question relates to how specific does FOXA2 selectively recruit Tet3 to P2. The authors made argument that the suggested experiment of TET3 ChIP-seq with FOXA2 knockdown is beyond the scope of our current study. I think the authors made the point and I am not going to hold the paper for this one particular experiment.

Reviewer #3 (Remarks to the Author):

In this manuscript, the authors determined the regulatory effect of TET3 on the expression of the P2 isoform of Hnf4 α , which is predominantly expressed in the fetal liver. They found that TET3 promoted hepatic glucose production (HGP), and played an important role in the upregulated HGP observed in type 2 diabetes animal

models such as HFD and ob/ob mice. With the experiments added, the authors showed that TET3 is preferentially recruited to the P2 promoter via its interaction with FOXA2 upon glucagon stimulation. The increased P2 HNF4 α proteins would then increase the gluconeogenic genes and promote glucose output.